# Design and Manufacture of Bone Cements Based on Calcium Sulfate Hemihydrate and Mg, Sr-Doped Bioactive Glass

**DOI:** 10.3390/biomedicines11102833

**Published:** 2023-10-18

**Authors:** Nazanin Moazeni, Saeed Hesaraki, Aliasghar Behnamghader, Javad Esmaeilzadeh, Gorka Orive, Alireza Dolatshahi-Pirouz, Shokoufeh Borhan

**Affiliations:** 1Nanotechnology and Advanced Materials Department, Materials and Energy Research Center, Karaj 31779-83634, Alborz, Iran; nazanin.moazeni@gmail.com (N.M.); a-behnamghader@merc.ac.ir (A.B.); 2Department of Materials and Chemical Engineering, Esfarayen University of Technology, Esfarayen 96619-98195, North Khorasan, Iran; j_es65@yahoo.com; 3NanoBioCel Research Group, School of Pharmacy, University of the Basque Country (UPV/EHU), 01006 Vitoria-Gasteiz, Spain; gorka.orive@ehu.eus; 4Biomedical Research Networking Centre in Bioengineering, Biomaterials and Nanomedicine (CIBER-BBN), Institute of Health Carlos III, 28029 Madrid, Spain; 5Department of Health Technology, Technical University of Denmark, 2800 Lyngby, Denmark; aldo@dtu.dk; 6Department of Materials, Chemical and Polymer Engineering, Buein Zahra Technical University, Buein Zahra 34518-66391, Qazvin, Iran; shkfborhan@bzte.ac.ir

**Keywords:** calcium sulfate, bioactive glass, rheology, bone cement, chitosan, Sr-doped biomaterials

## Abstract

In the present study, a novel composite bone cement based on calcium sulfate hemihydrate (CSH) and Mg, Sr-containing bioactive glass (BG) as solid phase, and solution of chitosan as liquid phase were developed. The phase composition, morphology, setting time, injectability, viscosity, and cellular responses of the composites with various contents of BG (0, 10, 20, and 30 wt.%) were investigated. The pure calcium sulfate cement was set at approximately 180 min, whereas the setting time was drastically decreased to 6 min by replacing 30 wt.% glass powder for CSH in the cement solid phase. BG changed the microscopic morphology of the set cement and decreased the size and compaction of the precipitated gypsum phase. Replacing the CSH phase with BG increased injection force of the produced cement; however, all the cements were injected at a nearly constant force, lower than 20 N. The viscosity measurements in oscillatory mode determined the shear-thinning behavior of the pastes. Although the viscosity of the pastes increased with increasing BG content, it was influenced by the frequency extent. Pure calcium sulfate cement exhibited some transient cytotoxicity on human-derived bone mesenchymal stem cells and it was compensated by introducing BG phase. Moreover, BG improved the cell proliferation and mineralization of extracellular matrix as shown by calcein measurements. The results indicate the injectable composite cement comprising 70 wt.% CSH and 30 wt.% Mg, Sr-doped BG has better setting, mechanical and cellular behaviors and hence, is a potential candidate for bone repair, however more animal and human clinical evaluations are essential.

## 1. Introduction

It is the standard procedure for surgeons to use donor bone or synthetic materials to fill up osseous defects left by procedures like arthroplasty, bone tumor removal, infection, fracture, or even trauma. Alternatives to allografts and autografts have been available for many decades in the form of synthetic polymers, ceramics, and polymer/ceramics composites. These bone substitutes and fillers are available in different shapes, including bulk pieces, injectable non-cement pastes, and moldable cement-type formulations [1,2,3,4,5,6].

Bone substitutes cements (BSCs) are composed of powder and appropriate solution phases and by mixing, they give a shapeable and self-setting paste. Therefore, these BSC biomaterials must have good rheological and injectable properties prior to self-setting to allow the surgeon to deliver bone cements paste through a needle or cannula into bone defects with irregular shapes and without proper access during the surgical process.

Calcium sulfate (CS), tri-calcium phosphate, and hydroxyapatite are some typical forms of bone fillers used therapeutically [7,8]. Among bone cements, calcium sulfate cement is known as a well-tolerated, fast setting, rapidly and completely bioresorbable material which not only is able to be used as bone filler, but can also be used as local delivery media of therapeutic agents [9]. Despite the favorable properties, calcium sulfate suffers from some weak points such as the rapid resorption rate and disability in chemical bonding with adjacent tissues (bioacivity) and weak mechanical strength [10,11]. Allthe above-mentioned points remarkably limit the clinical applications of CS cements. To overcome the above-mentioned challenges, researchers have engineered calcium sulfate composite with more suitable properties using different types of additives such as natural polymers (e.g., sodium alginate, gelatin, collagen) and bioactive inorganic materials such as hydroxyapatite and bioactive glass (BG) [12,13,14,15,16]. It has been shown that incorporation of some polymers into CS cement matrix could enhance the injectability, cohesion, and mechanical properties [17].

Beside the polymeric additives, the presence of inorganic nanoparticles such as hydroxyapatite and BGs can offer the physical and mechanical properties and regulate the degradation rate and the bioactivity of CS cements, owing to their dissolution products, which act as triggering osteoprogenitor cells at the genetic level. In one study [18], the addition of titanium-doped nano-hydroxyapatite in CS-based composites enhanced the setting time, mechanical strength, and bioactivity. Another study examined cement pastes composed of three biopolymers of gelatin/alginate/chondroitin sulfate and α-CSH and calcium-deficient hydroxyapatite (CDHA) [17]. Their results demonstrated the composite pastes exhibited optimal rheological behaviors, good handling properties as well as appropriate anti-washout characteristics. BGs in combination with various polymers have also been introduced into the CS cement; the biological, physicochemical, and mechanical features of the acquired cement pastes were investigated [9,17,19]. BG can form a chemical bond with living tissues and promotes bone formation in areas distant from the implant site [20,21].

Chitosan is an appropriate natural polymer with distinctive advantages that can be incorporated with cements to provide adequate physical and biological properties. It is hemostatic, fungistatic, and biodegradable, and exhibits antitumor, immunoadjuvant characteristics. It can bind to human and microbial cells, and accelerates the generation of osteoblast, which is responsible for bone formation [22]. In addition, chitosan shows antimicrobial property against Gram-positive and negative bacteria. The antibacterial activity exhibits a considerable dependence on both the cationic charge and the molecular weight [23]. These diverse features of chitosan make it easy for them to be used in scaffolds, nanoparticles, beads, microparticles, nanofibers, membranes, and other forms [22].

One of the important characteristics of CS-based cements is injectability. Injectability provides non-invasive transfer of cementitious material into the bone cavities through a needle or cannula, followed by quick in situ setting of the filling paste [24]. Since flow behavior is a key factor in injectable systems, the rheological characteristics of the cement pastes should be carefully assessed. Zima et al. [25] conducted research on the rheology of CS cements, emphasizing the influence of the ion presence on multiple inflections in the viscosity-time diagram.

In the present study, a novel bone cement based on calcium sulfate, Mg, Sr-containing bioactive glass, and chitosan was developed. A series of cements were prepared by mixing various amounts of sol-gel derived Mg, Sr-doped BG particles, and α-CS as powder phase and chitosan solution as liquid phase. The rheological behavior, injectability, and the correlation between injection force and viscosity were determined. The phase composition, morphology, setting time, and finally cellular cytocompatibility were also assessed.

## 2. Materials and Methods

### 2.1. Initial Powders Preparation

Calcium sulfate hemihydrate, CSH (CaSO_4_·0.5H_2_O, CAS-No. 10034-76-1, product No. 12090) was purchased from Merck. Citric acid and medium molecular weight chitosan (CAS-No. 9012-76-4, product No. 448877), C_12_H_24_N_2_O_9_, with M_W_: 200,000 g/mol, Deacetylation ≥ 75%, and purity > 98%, were prepared from Sigma-Aldrich, Taufkirchen, Germany. Mg, Sr-doped bioactive glass (BG) was also synthesized according to the previously described method [26] in 64SiO_2_-30CaO-5P_2_O_5_ system, in which the concentration of Sr and Mg dopants was 0.5 wt.% of CaO component.

### 2.2. Cement Pastes Preparation

The CS–BG cements were made by a combination of calcium sulfate hemihydrate and BG powders in various mass ratios, and chitosan and citric acid solution with a powder to liquid ratio of 1.5. Table 1 shows the composition of several CS–BG cements made at a constant P/L of 1.5 g/mL.

### 2.3. Setting Time, Porosity and Compressive Strength

The initial and final setting times of the CS–BG pastes were determined using a standard test with Gillmore needles, as described in ASTM C266–99 [27].

To determine the compressive strength, the cylindrical specimens were fabricated with a diameter of 6 mm and a height of 12 mm. At 24 h after setting, the specimens were transferred to a mechanical testing device (SANTAM STM-20) and the compression tests were conducted with at a loading speed of 1 mm/min.

The total porosity (P) of the cements was calculated on the cylindrical specimens, Archimedes method, where ethanol was used as medium [28].

### 2.4. Phase Composition

The phase composition of as-set cements was determined using X-ray diffractometry, XRD, (Philips PW 3710, Amsterdam, Netherlands) with Cu-Kα radiation and a 1.54050 nm X-ray wavelength, which was operated at 40 kV and 40 mA with a step size of 0.02 and a count duration of 2 s/step.

### 2.5. SEM Observation

The morphological and chemical characteristics of CS–BG cements were investigated using scanning electron microscope (SEM) (TESCAN-XMU, VEGA Ⅱ, Brno–Kohoutovice, Czech Republic) equipped with an energy dispersive X-ray spectroscopy (EDS) analysis unit. The map of S, Si, and N elements also determined the elemental image analysis. Moreover, the particle size analysis of the microstructure was performed using Image J software version 1.0.8_112 and the graphs were drawn by ORIGIN PRO 2022.

### 2.6. Injectability

An extrusion test was used to assess the paste injectability [29]. After 1-min mixing procedure, the obtained paste was placed into a commercial 3 mL syringe (AvaPezeshk Co., Tehran, Iran) with cannula length and tip diameter of 10 mm and 2 mm, respectively. The paste was exerted in syringe tube (the length of filled paste in the tube was 30 mm) and extruded from the syringe (held rigidly in a fixture) at predetermined times after the mixing process was completed by applying force using a universal testing machine (STM-20, Santam Co., Tehran, Iran) at a crosshead speed of 30 mm/min. The injection curve was plotted as injection force vs. plunger displacement. The injectability coefficient (I) was calculated as
I = [(M0 − M)/M0] × 100 (1)
where M0 is the initial mass of the composite in the syringe and M is the mass left inside the syringe after extrusion.

### 2.7. The Viscosity Measurements

The Rheometer (Anton Paar, Physica MC-R301, Graz, Austria) with plate–plate measuring geometry was used to examine the rheological properties of calcium sulfate-bioactive glass pastes (plate diameter of 25 mm). For this measurement, a homogeneous paste was made during 1 min and the resultant paste was subsequently applied to the lower plate’s center at predetermined intervals. The rheological test began 2 min after mixing the powder and liquid phases at both oscillatory and rotation modes.

A modest (low) amplitude sinusoidal oscillation strain was firstly applied to the paste in a dynamic oscillation test. The complex viscosity can be stated using the following formula due to the phase difference between two sinusoidal waves.
G* = G′ + iG″(2)
ղ* = ղ′ − iղ″(3)
ղ′ = G″/ω (4)
ղ″ = G′/ω (5)
where ω is the frequency in rads^−1^, G′ is shear storage modulus, G″ is shear loss modulus, G* is complex shear modulus, ղ″ is out-of-phase viscosity, ղ′ is dynamic (absolute) viscosity, and ղ* is complex viscosity. The real part of the complex viscosity (dynamic viscosity) (ղ′) is a criterion of force required to make a fluid flow at a certain speed [30,31]. Firstly, strain sweep mode measurement was performed to identify the linear viscoelastic region (LVR). Then, the dynamic frequency sweep test was performed while maintaining constant strain (1%) and temperature (25 °C).

In rotatory mode, the viscosity–time curve was also drawn at a constant shear rate of 1 s^−1^ to identify the variation of the paste viscosity as a function of time.

### 2.8. The Cell Studies

#### 2.8.1. Cell Survival, Viability, and Growth

For in vitro cellular studies, the samples were immersed in an antibiotic solution for a duration of 30 min; afterwards, they were rinsed in phosphate-buffered saline (PBS), and subjected to ultraviolet irradiation (UV) for a period of 15 min.

The cytotoxic effect of cements on human bone mesenchymal stem cells (hMSCs) was studied by 3-(4,5-dimethylthiazol-2)-2,5-diphenyltetrazolium bromide assay (MTT; Sigma-Aldrich, Taufkirchen, Germany). The cells were seeded in 100 μL of the medium at a density of 10^4^ cells/well in 96-well culture plate and were incubated at 37 °C in a 95% humidified atmosphere of 95% air and 5% CO_2_, for 24 h. Subsequently, spherical specimens (with the same shape and surface area) were positioned in the cell culture medium and incubated for 24, 48, and 72 h. The sample-free culture plate was used as control group. After incubation, the medium was evacuated and 100 μL of serum free MTT-containing culture medium (0.5 mg/mL) was added to each well, followed by a 4-h incubation at 37 °C. Finally, 100 μL of dimethyl sulfoxide (Sigma, Germany) was added to each well, and cell viability was determined using a microplate reader (BioTek ELx800, Winooski, VT, USA) at 570 nm. The percentage of viable cells (compared to the control group) were evaluated and reported.

#### 2.8.2. DAPI Staining

DAPI (4′,6-diamidino-2-phenylindole) is a fluorescent stain that binds strongly to adenine–thymine rich regions in DNA and allows for observing the health or death of the cell. For this purpose, the hBMSCs was cultured on the cements, and on days of 1, 3, and 7, it was taken out of the incubator for staining, the complete culture medium was removed and washed once, then fixed with 4% (*w*/*v*) paraformaldehyde for 30 min. The paraformaldehyde was removed, the DAPI solution (D9542-1MG SIGMA) with a concentration of 300 nM was poured on the cements and after 5 min, it was observed under the fluorescent microscope NIB-100F (NOVEL Co, Ningbo, China).

#### 2.8.3. Calcein Staining

Cells were cultured on the cement according to the standard procedure. On the desired day of study (days 14 and 21), cements containing cells were washed to the assay with 1000 µL of a phosphate buffer to remove any serum esterase activity that may be present in the growth media. Calcein powder (L468415 MERCK) was dissolved in DMSO and a working solution with a final concentration of 2 μM was prepared. Next, about 200 microliters was added to the samples, so it was covered and incubated at temperature 25 °C for 30 min. After the passage of time, it was observed with a fluorescent microscope NIB-100F (NOVEL Co. Ningbo, China Co.)

### 2.9. Statistical Analysis

The data were collected from at least three separate experiments. The one-way ANOVA was performed for the statistical analysis, and *p* value < 0.05 was considered statistically significant (* *p* < 0.05, ** *p* < 0.01, *** *p* < 0.001).

## 3. Results

### 3.1. Phase Composition

Figure 1 illustrates the XRD patterns of various CS–BG cements as well as starting materials (chitosan, CSH and BG) presented for comparison. The x-ray diffraction patterns of chitosan and bioactive glass powder reveal characteristics of an amorphous phase. In the patterns of set cement CS100-BG0, the unreacted CSH is the predominant phase and some minor gypsum product is observed (PDF-Number 33-0311). The same phase composition can also be seen in the diffraction patterns of BG-containing cements, except an amorphous background seen due to the presence of BG phase. The amorphous-like background in the XRD patterns of these samples may also be related to the formation of amorphous chelate (complex) produced from the chemical reaction of Ca ions (delivered from BG and CSH) and citric acid and/or chitosan molecules.

### 3.2. Setting Time

The initial and final setting times, compressive strength and porosity of the cements are given in Table 2. From here, we can see that while pure calcium sulfate hemihydrate combined with pure distilled water sets at about 35 min, the chitosan-containing calcium sulfate cement exhibits a very long setting time. The C100-BG0 is also weak in terms of compressive strength. A significant decrease in setting time was observed when BG was included into the composition. Here, we observed a significant difference between the setting times of the various compositions, especially for CS70-BG30. The compressive strength is drastically improved in high contents of BG. Also, the porosity is in the range of 30–60% and except for CS90-BG10, adding BG decreases the total pore content.

### 3.3. SEM Observation

Microstructures of investigated CS–BG cements are shown in Figure 2 and Figure 3. Figure 2 (A1–D3) depicts the SEM images and related EDX analysis of CS–BG cements. The results of particle size distribution performed by Image J software are shown in the inset of each related image. CS100-BG0 has porous structure morphology consisting of interlocking hexagonal flake-like crystals. Ca, S, and O elements are found in the related EDXA image of CS100-BG0. A mono-modal particle size distribution with average size of 7.9 µm is observed. By adding BG to calcium sulfate, the following changes in the microstructure are observed: (i) Fine glass particles embedded within a monolithic-shaped phase that covered the large block-like calcium sulfate crystals are seen; (ii) The morphology and shape of flakes changes from regular hexagonal to relatively shapeless blocks particles (particularly for CS90-BG10 and CS80-BG20); (iii) In CS70-BG30, a flake-like morphology with reduced thickness and particle size is observed. The particle size distribution determines that when BG is added, a bimodal size distribution is found. In the authors’ opinion, the smaller average size relates to remaining BG that did not react with chitosan molecules, whereas the larger average size is attributed to calcium sulfate crystals. By increasing the BG content, the average size of calcium sulfate crystals decreases, so that in CS70-BG30, the bimodal graph tends to a mono-modal. In the EDXA patterns of BG-added CS, in addition to Ca, S, and O, the P and Si elements are also found. The peaks of Sr overlap with P and Si, meanwhile the detection of Mg and Sr peaks is impossible because of the very low concentration of them in the glass composition. It should be noted that the EDXA is not a quantitative analysis and the content of the elements should not be judged by the intensity of the peaks.

Figure 3 depicts the SEM elemental mapping of CS–BG samples for elemental distribution. Blue spots indicate the presence of Si as the principal element in the BG structure. Green dots represent S (Sulfur) as representative of CaSO_4_, whereas yellow dots symbolize N (nitrogen) as ingredient of chitosan. The SEM of the surfaces of the samples (provided in the left side of the map images) is in agreement with those shown in Figure 2. The addition of BG to calcium sulfate alters the microstructure, and when the percentage of bioactive glass reaches 30 percent, the microstructure of calcium sulfate-bioactive glass changes generally. The surface of CS–BGs shows a significant difference when the BG content increased. In CS90-BG10 and CS80-BG20, large dark areas are observed and the Si distribution seems heterogeneous. It can be related to the inhomogeneity of the surface created by the large pores. In the case of CS70-BG30, with a higher percent of BG, the microstructure was changed and fine flakes along with the remaining CSH reactants are observed. Also, the SEM image reveal that the microstructure of the CS70-BG30 contains a finer particle, more reduced porosity, denser and smoother surfaces. Therefore, the distribution of blue dots seems more homogenous.

### 3.4. Injectability

Figure 4A shows the injection curves of cements with different contents of BG. At the beginning point of test, a sudden increase in extruding force is observed for all cements. This was also observed in other research. In a pilot test, the force-displacement diagram of empty syringe was checked. In this situation, no overshoot was observed at the start of the test. It determines that the syringe wall-plunger friction is negligible and the beginning overshoot corresponds to the force required to overcome the paste-syringe wall friction caused by hydraulic pressure inside the syringe [29,32]. After that, all the cements are injected with a nearly constant force, lower than 20 N. Overall, it is found that the composites with higher BG content exhibit higher injection force. No filter-pressing phenomenon is observed during the injection. It seems that the extrusion force decreased after 5 mm of displacement. It was found that the paste was coherent and slippery due the presence of chitosan. The authors suggest, when a volume of the paste is extruded, lower force is required to extrude the remaining paste. Thus, a decreased profile is seen for the extrusion by syringe displacement.

Since in the injectability test, the paste is extruded at a nearly constant rate, and the force required for the extrusion is evaluated, it may be possible to create an equivalence between the real part of the viscosity (ղ’) and the force required for injection. It means that ղ’ can be considered a criterion of the force required for injection.

Figure 4B shows the real part of the complex viscosity (ղ’) of various cements as a function of ω calculated from Equation (4). In other words, there is a direct correlation between the injection force and (ղ’). It can be seen that the starting points of ղ’ increase with introducing considerable amount of BG, which is consistent with the injectability curve of cements. It means that a higher injection force is required. Moreover, at low frequencies range (ω = 1 − 8 s^−1^), with increasing ω, a sharp decrease in the real part of viscosity is observed (especially CS70-BG30) and the force required the injection to be continued decreases.

### 3.5. Viscosity

Figure 5 shows the complex viscosity (ղ*) of the paste as a function of ω. The complex viscosity of cements with higher percent of bioactive glass is more sensitive to angular frequency.

The angular frequency value of 10 s^−1^ is the intersection point of ղ*-ω curves. In other words, at ω < 10 s^−1^, cements with higher BG content have a higher ղ* value and the cement CS70-BG30 exhibits the highest complex viscosity. At frequencies higher than 10 s^−1^, the viscosity behaves completely different and samples with more bioactive glass content have a lower viscosity. In this case, the highest viscosity relates to the glass-free cement. In fact, more internal structures can be formed in pastes containing bioactive glass, and when the frequency exceeds 10 s^−1^, these structures are broken leading to a sudden decrease of viscosity. In the case of these cements, with an increase in frequency (more than 100 s^−1^), the viscosity increases again and shear thickening behavior is observed. It means that the destroyed internal structures (links) are reformed due to the intensive stresses.

Figure 6A depicts the result of viscosity in rotatory mode. The change in viscosity of the cements as a function of time is shown. Figure 6B illustrates the viscosity–time curves of three cements CS100-BG0, CS90-BG10, and CS80-BG20 to clarify the viscosity variations in more details. There was a considerable difference between viscosity and the viscosity changes of CS70-BG30 and other cements. For CS100-BG0, CS90-BG10, and CS80-BG20, a gradual increase in the viscosity is observed up to 1200 s. In cement CS70-BG30, the viscosity is gradually increased to approximately 294 (Pa.s); by 300 s, an increase is observed by which the setting reaction is initiated and the viscosity reaches to 29,700 (Pa.s). At t = 550 s (corresponds to initial setting time), relatively reached a plateau up to 930 s. Finally, a sharp progress in viscosity begins at time 930 s, which can be corresponded to the final setting time.

According to the viscosity–time diagram of CS70-BG30, the final setting time is much shorter than that estimated by Gillmore needles. In other words, the Gillmore needles provide a more limited definition of final setting time. Sarda et al. who evaluated the viscosity–time diagram of α-tricalcium phosphate cement [33] observed the same results. Due to the relatively long initial and final setting times of other cements (CS100-BG0, CS90-BG10, and CS80-BG20), the above-mentioned changes are not seen in their viscosity–time diagrams.

At time range of 950–1000 s, a gap is observed in the viscosity curve of cement CS70-BG30. It is related to the intensive fluctuations that cannot be shown in logarithmic scales. The fluctuations are also observed in cements with lower amounts of BG (Figure 6B) in which the intensity of fluctuations increases with increasing the BG content. As observed by other authors [19,34], the presence of these fluctuations can be due to the formation and destruction of local structures as a result of (I) CSD formation as well as (II) chelate creation by the chitosan and alkaline earth ions (such as Ca, Mg, and Sr).

### 3.6. Cell Studies

#### 3.6.1. Cell Viability and DAPI Staining

Figure 7 displays cell proliferation of hBMSCs cultured on CS–BG cements and cytotoxicity of cements against the hBMSCs cell was determined by an MTT bioassay.

To determine the proliferation rate of hBMSCs on the various CS–BG cements, the number of viable cells after 4 h of cell culture was determined and used as a standard. The number of cells grew with culture time, and the rate of cell proliferation varied according to the percentage of bioactive glass composition. In general, samples with more bioactive glass demonstrated greater cell viability than CS100-BG0, which is in agreement with other studies [20,35]. On the sample with the highest BG percent, more cell division was observed. It has been stated that Mg (Magnesium) and Sr (Strontium) as therapeutic ions in bioactive glass have biological properties. Sr improves mechanical stability and increases in vitro bioactivity and bone cell adhesion [36].

Fluorescent microscopy was used to examine the distributions, viability, and morphology of hBMSCs cells in CS–BG cements after DAPI staining. After 1, 3, and 7 days of cell culture, all alive cells were blue, as illustrated in Figure 8. More viable cells were observed on the CS70-BG30 cement with the highest bioactive glass component, and the cells distributed on it more evenly than pure CS cement.

#### 3.6.2. Calcein Staining

Figure 9 focuses on mineralization, which is one of the most important late osteogenic markers. Figure 9 shows the calcium deposits produced by hBMSCs on CS–BG cements. From the images, we were able to identify green dots corresponding the formation of calcein/calcium complexes. To analyze the mineralization of CS–BG cements, the culture media was supplemented with calcein, a calcium-binding fluorescent dye. In mineralized ECM, calcein can bind to Ca^2+^, leading to bright green stains. Two and three weeks after culturing, the calcein fluorescence intensities in the mineralized matrix of CS, and CS–BG cements were seen. CS70-BG30 exhibited a more intense and widespread fluorescence than CS100-BG0 in both incubation times.

## 4. Discussion

The setting reaction of pure CSH is assigned to the formation of gypsum crystals due to the hydrolysis of CSH, as described by several authors [19,37]. In brief, the setting reaction of calcium sulfate cement consists of the following important events [24,38]: (i) Partial dissolution of CSH in the water phase of the formed workable suspension; (II) Saturation of the aqua solution with calcium sulfate dihydrate (CSD) around the CSH particles; (iii) Precipitation of CSD; (iv) Final dissolution of the remaining CSH as the saturation degree decrease due to the CSD precipitation. The growth of precipitated CSD nuclei progresses through the dissolution–precipitation phenomenon and finally, the cement hardens because of the entanglement of grown CSD (gypsum) crystals.

In calcium sulfate cements investigated in this study, citric acid and chitosan were used in the cement liquid phase. It has been reported that citric acid has an impact on the mechanical properties, kinetics, and biocompatibility of calcium salt-based cements, in a concentration-dependent manner [39,40]. Citric acid was also selected to provide an acidic pH to dissolve chitosan. Chitosan was employed for its unique properties mentioned in the previous section. The results demonstrated that these additives prolong the setting phenomenon of calcium sulfate. The effect can be related to the presence of citric acid, chitosan, or both of these organic additives. To clarify this point, calcium sulfate cements made of pure CSH as powder and various liquids including pure distilled water and citric acid solution were also prepared. Based on the results, the cement sets at about 35 min when pure distilled water was employed, whereas in the case of citric acid solution, the setting reaction lasts more than 240 min. Yokoyama et al. [40] demonstrated that the carboxyl and hydroxyl groups of citric acid molecules react with Ca ions (introduced from CS and BG) to form calcium citrate molecule. The authors suggest that the Ca^2+^ ions are consumed to form calcium citrate and hence, the supersaturation with respect to CSD does not achieve. Moreover, the formed citrate molecules may be absorbed on the surfaces of CSD nuclei and restricts its growth and entanglement. Zhao et al., who investigated the inhibition of calcium sulfate scale by poly (citric acid), observed the same results [41]. The Ca ion absorption phenomenon can also occur by the chitosan molecules. Authors suggest that chitosan, which possesses several chemical agents (reactive amino groups and hydroxyl groups), absorbs the ions present in suspension and retards the supersaturation of the solution with respect to CSD. Hence, the capture of Ca ions by both citric acid and chitosan molecules and absorption on the CSD nuclei surfaces would result in a more prolonged setting time (more than 600 min). Zima et al. [25] stated that chitosan molecules form a polymeric layer around the calcium sulfate crystals, resulting in a delayed hydration process and hence retards the main setting reaction (gypsum precipitation). As confirmed in the XRD Figures, the peaks associated with CSH can still be seen in all composites. In the present study, however, the setting time improves by introducing bioactive glass particles. In the authors’ opinion, the secondary setting mechanism is responsible for this phenomenon. It is the formation of an organic–inorganic complex network through the chelation of metallic ions (such as Ca^2+^, Sr^2+^, and Mg^2+^) released from the surfaces of reactive glass particles with carboxyl, hydroxyl, and amino groups of organic molecules. Based on the literature, it is well acknowledged that amine groups can react quite well with metal ions, while the hydroxyl groups can contribute to sorption—something which typically occurs at neutral or slightly acidic pH values [42,43]. According to the literature, chitosan not only forms complexes with transitional ions but also creates complex with alkaline and alkaline-earth ions (e.g., Mg^2+^, Ca^2+^, Sr^2+^) [16,19,41]. Various degrees of chelate formation in the presence of calcium phosphate cement and carboxylic acids has also been demonstrated in the previous literature [38].

In this study, the effect of bioactive glass particles on compressive strength and porosity was determined. The results showed that high content of BG improved mechanical strength. Compared to hydroxyapatite (which has been added to calcium sulfate in other studies [44]), the addition of bioglass to CSH has a different effect on both compressive strength and setting time. In previous reports, introducing hydroxyapatite into CSH retarded setting time and sharply decreased the compressive strength. This can be due to the reaction of BG particles with chitosan molecules, which can create a secondary setting reaction and monolithic chelate (as shown in SEM images), resulting in increasing compressive strength. In other words, hydroxyapatite only acts as an inert filler that reduces the concentration of calcium sulfate hemihydrate reactant and causes a longer setting time and a decreased strength. In this study, the increase in the strength of the cements containing bioactive glass can also be due to the decrease in the percentage of porosity and the size of the crystals (as seen in the SEM images of Figure 2). Compared to other similar works, the results show that the CSH/BG cements of this study were highly porous composites. It can be the main reason for the lower compressive strength of cements of this study compared to previously reported work [15,21]; however, the composition, particle size, the solid to liquid ratio, and preparation method are the main determining parameters. Although the cements exhibit poor mechanical strength, the values are within the range of cancellous bone (0.036 to 2.945 MPa) and hence, they can be used for non-load bearing application.

Viscosity is another important intrinsic character of a cement. A functional injectable bone cement paste must have a viscosity lower than 1000 Pa.s. Extremely viscous pastes (viscosities greater than 1000 Pa.s) need a high injection force, which is challenging to be obtained with manually operated syringes/applicators. Very low viscosity may also result in extravasation of the loose paste from the surgical site into the surrounding tissues [30]. Accordingly, the CS-BG formulations have adequate viscosity, meaning that the cement pastes are expected to be extruded throughout the entire plunger travel distance.

Another goal of this study was to regulate cytotoxic effect of calcium sulfate and improve its osteogenic capacity. The distributions, viability, and morphology of hBMSCs cells on CS–BG cements were determined by DAPI staining. After different periods, all alive cells were detected. The percentage of bioactive glass was responsible for the variation in the number of living cells and cell spreading. More viable cells were observed on the composites contained the highest bioactive glass (e.g., CS70-BG30 cement), and the cells distributed on composites were more than pure CS cement which is in agreement with other studies [45]. A transient cytotoxicity of pure calcium sulfate has been reported by some researchers [46]. In this study, the same results were achieved that confirm other studies. Replacement of a fully biocompatible bioactive glass phase with a partially cytotoxic calcium sulfate phase improves the cell viability; meanwhile, the role of Si, Mg, and Sr ions in proliferation of hBMSCs should not be ignored [22,34]. Concerning the osteogenic capacity of the composite, it should be noted that hBMSC osteogenic differentiation comprises three major stages: (1) proliferation; (2) extracellular matrix (ECM) secretion; (3) mineralization. Each stage is distinguished by a unique set of markers that make it possible to track its progression over time [47]. Mineralization of the extracellular matrix is regarded as a sign for the ultimate stage of osteogenic differentiation. In this process, mineral deposits reach a maximum value after about 2 to 3 weeks of culture [48]. In this study, the calcein staining was used to assess the mineralized extracellular matrix of subcultured cells. After 14 and 21 days of culture, calcein incorporated into matrix mineralization was observed and visualized using a fluorescence microscope, indicating the efficacy of calcium sulfate/bioglass composites.

Overall, the authors state that the addition of Sr, Mg-containing bioactive glass to the calcium sulfate enhances biocompatibility and hBMSC behavior (viability, proliferation, and differentiation) resulting in a composite with improved biofunctionality. The cement can be used for orthopedic application including regeneration of lost bone with critical size defects. The composite was designed as a solid–liquid product and must be sterilized before the clinical use. The employed sterilization method should not deteriorate the physical, mechanical, and structural properties. Thus, the authors suggest the powder phase of the product can be sterilized by gamma irradiation or ethylene oxide and the liquid phase can be sterilized by standard filtration methods (which is suitable for liquids).

## 5. Conclusions

In this study, CS–BG bone cements have been prepared by incorporating calcium sulfate and bioactive glass powders, in various weight ratios, with a chitosan solution as liquid. The addition of bioactive glass to calcium sulfate alters the morphology of precipitated gypsum, and the microstructure changing depends on the concentration of glass phase. The setting time of pure calcium sulfate cement with a chitosan solution as liquid phase decreases by introducing Mg, Sr-containing bioactive glass phase, in concentration-dependent manner. Adding BG to CS cement decreases the total porosity and improves the mechanical strength. Cement with higher bioactive glass content requires higher force for a complete injection; however, the extrusion fore is lower than 20 N for all pastes. The complex viscosity of cements with higher percent of bioactive glass is more sensitive to angular frequency. In all composites, a reduction in complex viscosity (ղ*) is observed with increasing frequency which shows the shear thinning behavior of pastes. The addition of Mg, Sr-containing bioactive glass to calcium sulfate cement compensates its nearly transient cytotoxicity and improves the hBMSCs viability. The Mg, Sr bioactive glass incorporated with CS cement improves cell proliferation and mineralization of extracellular matrix. Although the combination of calcium sulfate and bioactive glass in some ratios (e.g., CS70-BG30) can produce suitable materials in the terms of injectability, biocompatibility and osteoconductivity, the main challenge is the relatively poor mechanical strength. Thus, the cement can be proposed for non-load bearing applications; however, more in vivo and human clinical investigations are required.

## Figures and Tables

**Figure 1 biomedicines-11-02833-f001:**
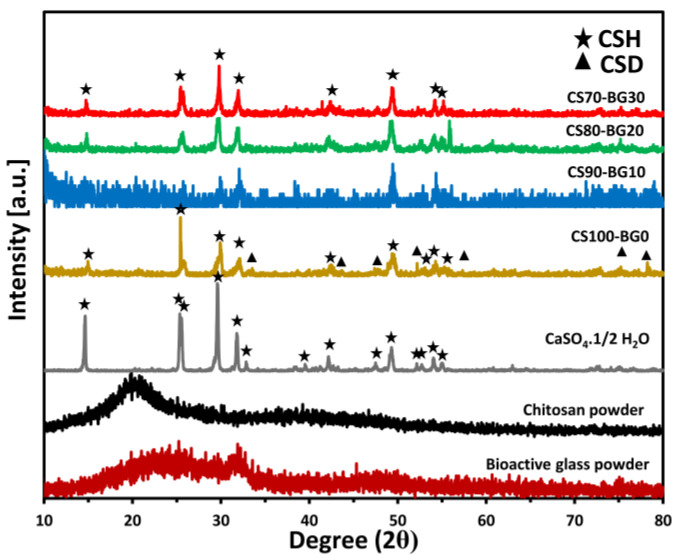
XRD pattern of initial powders and CS–BG cements.

**Figure 2 biomedicines-11-02833-f002:**
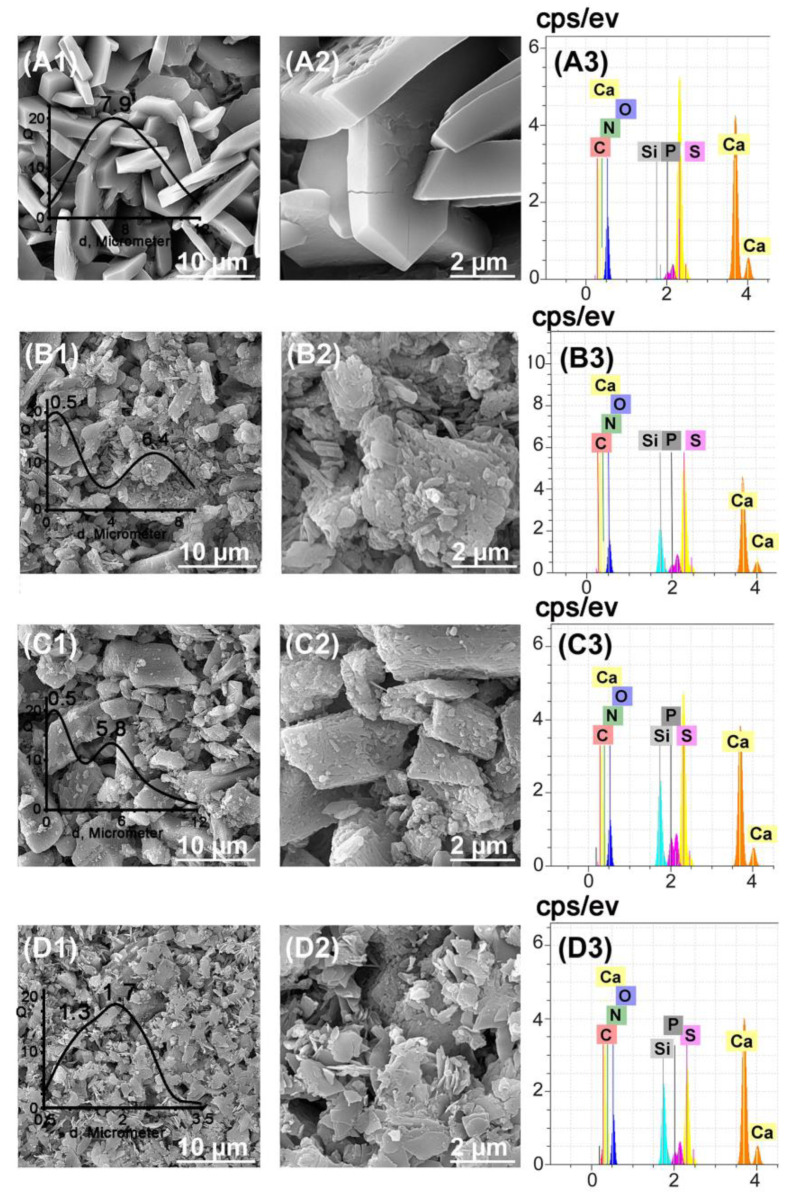
SEM and the corresponding EDS images of CS100-BG0 (**A1**–**A3**), CS90-BG10 (**B1**–**B3**), CS80-BG20 (**C1**–**C3**), and CS70-BG30 (**D1**–**D3**) cements.

**Figure 3 biomedicines-11-02833-f003:**
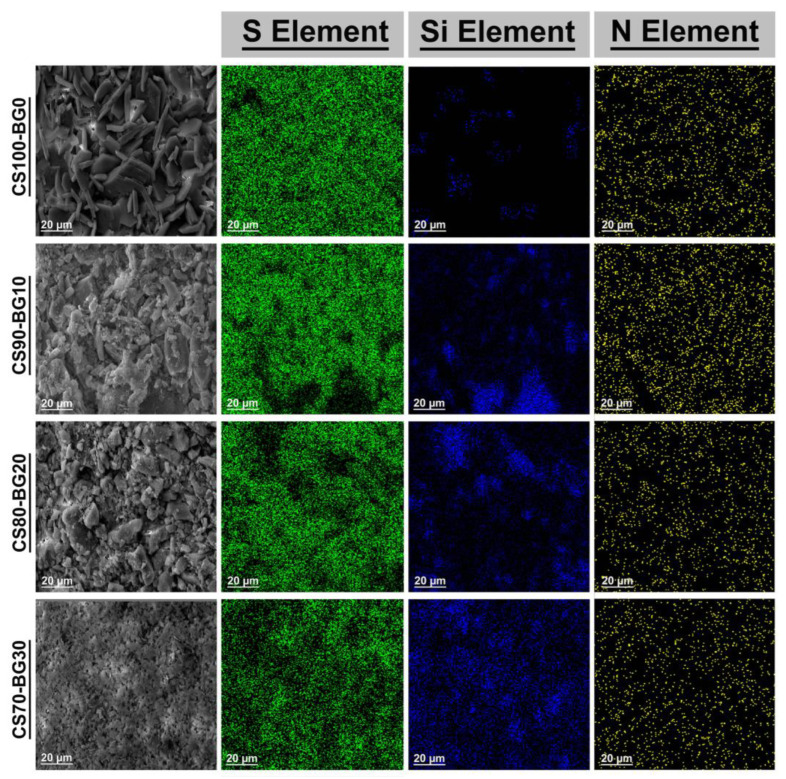
Surface morphology and elemental distribution of S, Si, and N.

**Figure 4 biomedicines-11-02833-f004:**
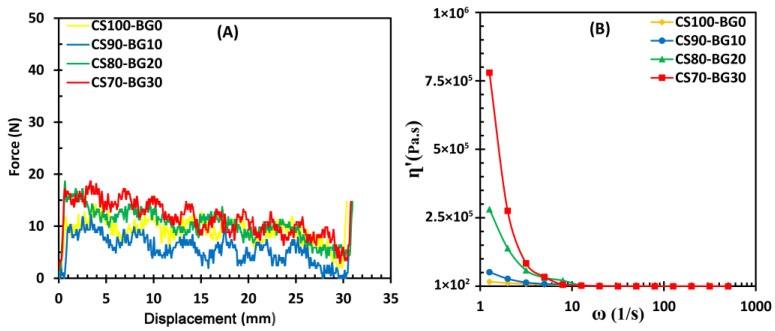
(**A**) The injection curves of different cements shown as applied force vs. plunger displacement. (**B**) The real part of the complex viscosity (ղ’) of various cements as a function of ω.

**Figure 5 biomedicines-11-02833-f005:**
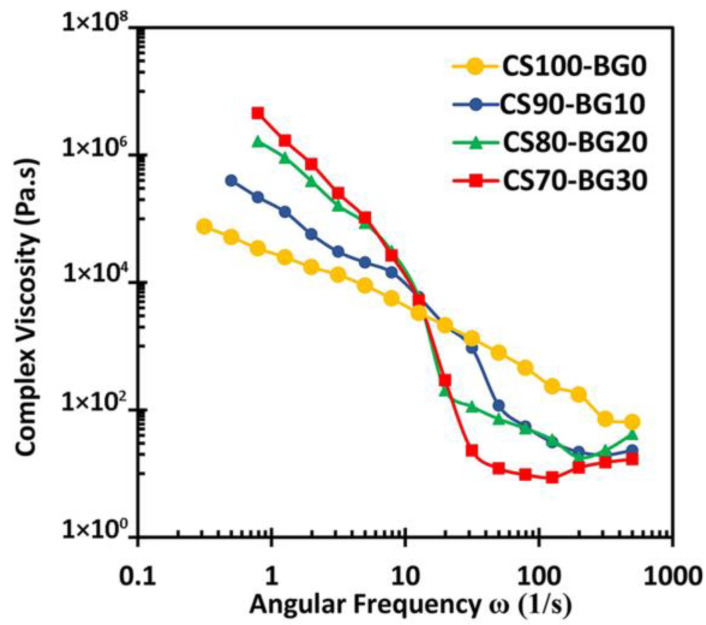
The complex viscosity of various pastes as a function of angular frequency (ω).

**Figure 6 biomedicines-11-02833-f006:**
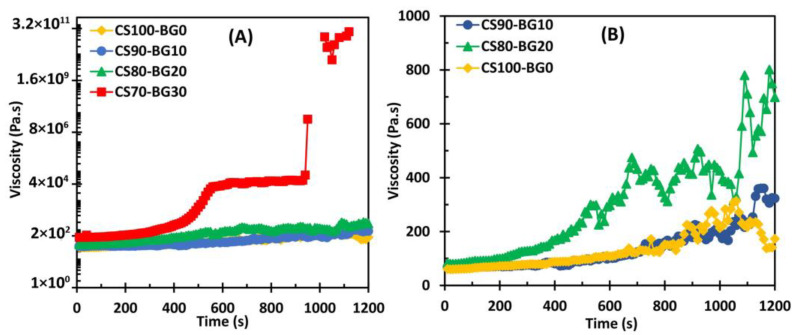
(**A**) The change in viscosity of the cements as a function of time. (**B**) Viscosity of CS100-BG0, CS90-BG10, and CS80-BG20 as a function of time.

**Figure 7 biomedicines-11-02833-f007:**
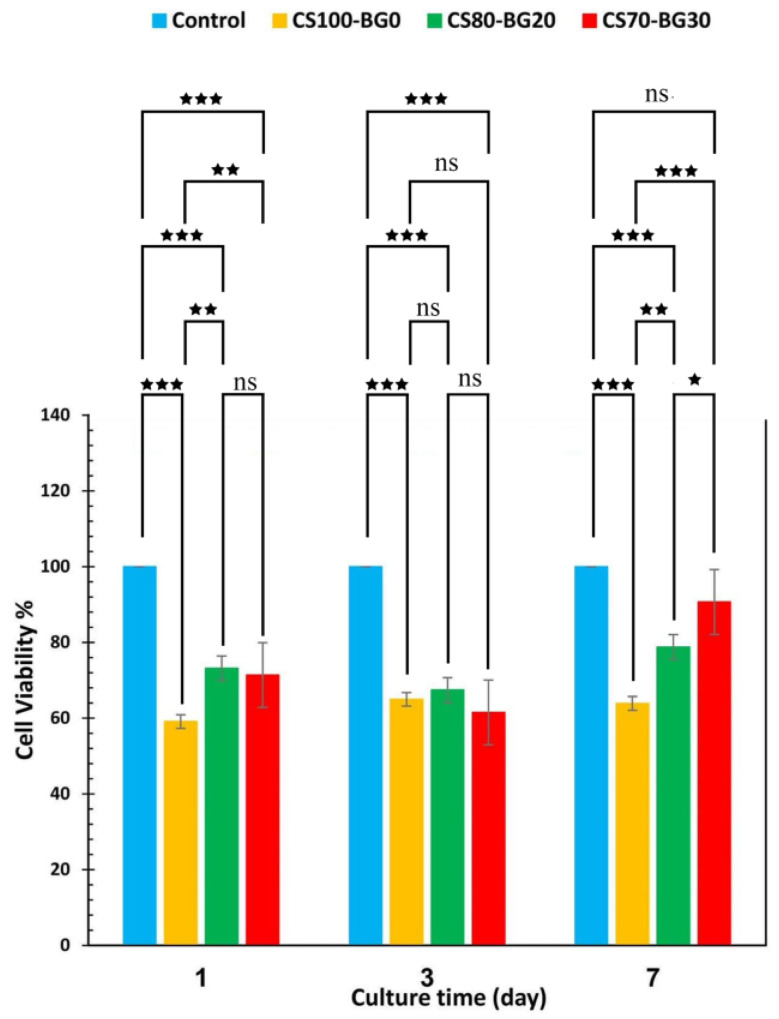
The proliferation of hBMSCs cultured on CS–BG cements (* *p* < 0.05, ** *p* < 0.01, *** *p* < 0.001, ns: non-significant).

**Figure 8 biomedicines-11-02833-f008:**
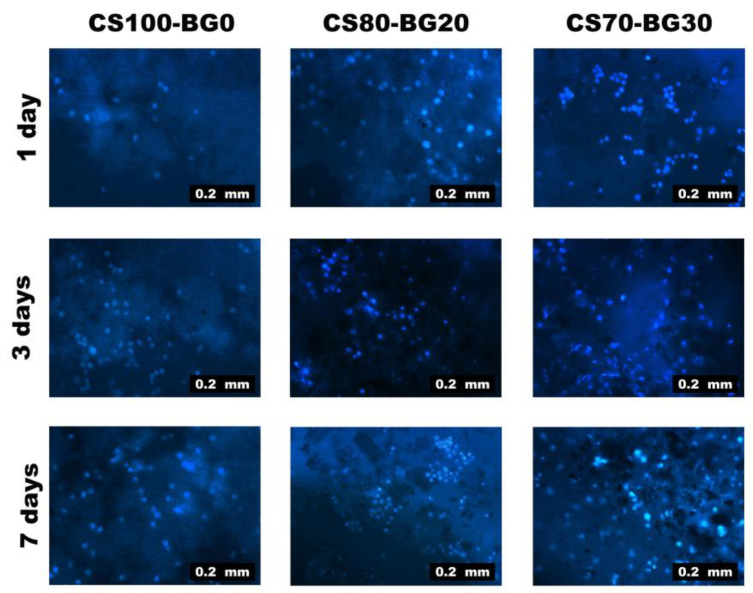
The viability, distribution, and morphology of hBMSCs on CS–BG cements after DAPI staining.

**Figure 9 biomedicines-11-02833-f009:**
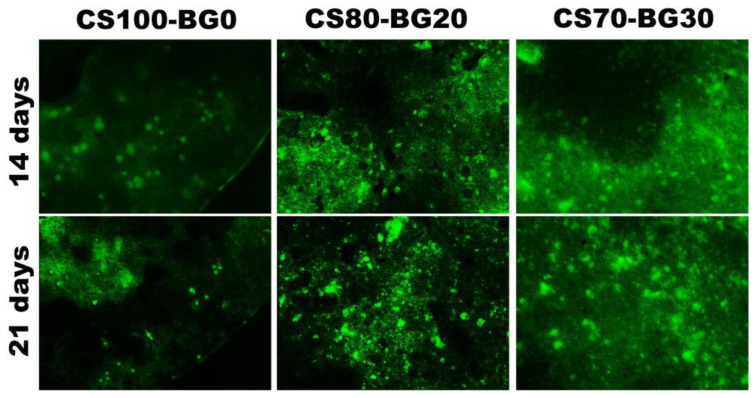
Calcein staining of hBMSCs cultured on cements on Days 14 and 21.

**Table 1 biomedicines-11-02833-t001:** Composition various CS–BG cements.

Code	Powder Phase		Liquid Phase
	CSH		Solution of 19.2% citric acid and 2% chitosan (*w*/*v*)
CS100-BG0	100	0	
CS90-BG10	90	10	
CS80-BG20	80	20	
CS70-BG30	70	30	

**Table 2 biomedicines-11-02833-t002:** Initial and final setting times, compressive strength, and total porosity of calcium sulfate cements containing different amounts of BG.

Sample	Initial Setting Time (min)	Final Setting Time (min)	Compressive Strength (MPa)	Porosity (%)
Cement made of pure CS and distilled water	25 ± 3	35 ± 3	-	-
Cement made of pure CS and citric acid	180 ± 15	240 ± 20	-	-
CS100-BG0	>300	>600	0.29 ± 0.02	50 ± 2
CS90-BG10	60 ± 5	150 ± 20	0.20 ± 0.07	61 ± 3
CS80-BG20	32 ± 5	120 ± 10	1.25 ± 0.11	38 ± 1
CS70-BG30	6 ± 1	20 ± 4	4.25 ± 0.85	36 ± 2

## Data Availability

Raw data were generated at MERC Institute. Derived data supporting the findings of this study are available from the author S. Hesaraki and/or Nazanin Moazeni on request.

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
