# Peer review of "Design and Manufacture of Bone Cements Based on Calcium Sulfate Hemihydrate and Mg, Sr-Doped Bioactive Glass"

_biomedicines, 2023, doi:10.3390/biomedicines11102833_

Round 1

Reviewer 1 Report

Regenerative medicine is a very important area and can significantly improve the efficiency of healthcare. Of course, the success of biomedicine is impossible without the intensive participation of materials science and polymer chemistry. The authors have a good feel for an important topic in modern materials science. The topic of the manuscript is relevant and original and occupies a specific gap in the field; the authors propose an original solution to an important issue using bone cements based on calcium sulfate hemihydrate, involving chitosan in the composite. This is a very correct decision, since chitosan is a biocompatible, biodegradable, non-degradable natural polymer and itself has a wide range of attractive biological properties. This compares this work favorably with other published data. The authors’ new data on the creation of a new composite cement are interesting and important for replenishing the knowledge base in the field of relationships between “composite composition and composite properties.”

The article is written very well, logically, well illustrated with pictures, tables, etc. Conclusions are fully consistent with the presented material. The experiment was carried out methodically correctly. All references are relevant.

I do, however, recommend a minor revision before publishing.

I kindly ask the authors

1) shorten the introduction, now it’s too verbose

2) indicate the average molecular weight of chitosan, as well as the polydispersity index. These are very important parameters on which the reproducibility of the experiment dramatically depends.

Author Response

The authors wish to acknowledge the reviewer 1 for consideration the manuscript and useful comments. Fortunately, the manuscript was reformed in accordance with your comments and the other reviewers and the changes are exerted in red-font. In more details: Reviewer 1 R1: 1) shorten the introduction, now it’s too verbose Author: The introduction section was deeply reformed. R1: 2) indicate the average molecular weight of chitosan, as well as the polydispersity index. These are very important parameters on which the reproducibility of the experiment dramatically depends. Author: In the materials and methods initial powder preparation, chitosan product number and CAS number and Mw were mentioned.

Reviewer 2 Report

This manuscript concerns the preparation and characterization of calcium sulfate cements based on calcium sulfate hemihydrate, chitosan, and bioactive glass. The idea of this manuscript is good, and it is very well presented. The manuscript is considered a welcome addition to the current literature, which attempts to develop and formulate new bone cements based on calcium sulfate hemihydrate and study their physico-chemical and biological properties. However, this paper contains some limitations that should be addressed, especially for rheology and injectability. My concerns and minor comments are outlined below:

1.       The abstract should be improved.

2.      Abstract: The authors said, "The rheological measurements in oscillatory mode determined the viscoelastic nature of the pastes, in which the elastic modulus overcomes the loss modulus, especially in the BG-containing cements." There are no claims for elastic and loss moduli in the manuscript.

3.      Since these bone cements are designated for orthopedic applications and more, the authors should explain which sterilization method is adequate for this type of biomaterial.

4.      How do the authors sterilize the samples before cell studies? This point is very important and must be highlighted.

5.      The authors confirm that the dynamic frequency sweep tests have been carried out within linear viscoelastic behavior. Please use the standard nomenclature for such a parameter: linear viscoelastic region (LVR).

6.      To homogenize the manuscript, please use elastic/viscous moduli or storage/loss moduli.

7.      The authors chose to use eta’ and eta’’ instead of G’ and G’’. Why this choice? It could be preferable to use G’G’’ as a function of frequency for each sample.

8.      Line 171: The dynamic shear modulus is G*, not G’. Further, the equation legend must include the signification of G’ and G’’ (i.e., storage modulus and loss modulus, respectively).

9.      Sub-section 2.8.1 (Cell Survival, viability, and growth): It is unclear how the cements were placed next to the cell and incubated for 24, 48, and 72 hours.

10.   Figure 4A: Replace "extension" with "displacement.".

11.    The authors should make a difference between room temperature and 25°C.

12.    The authors can add and use this bibliographic reference to highlight extrusion and the relationship between rheology and injectability for such biomaterials: https://doi.org/10.1007/s10856-012-4640-4

13.    Line 281: …the friction of the syringe surface and the paste. The authors should confirm if the existence of wall slips was checked in this case.

14.    Sub-Section "2.1. Initial powders preparation": the Mw of the chitosan should be specified.

15.    In sub-section "2.6. Injectability", the following information about the commercial 30 mL syringe should be specified: trademark, supplier, diameter, and length. In addition, there was no idea about the needle used in this study. If yes, the authors should provide the diameter and length of the needle. If not, the authors should explain why not.

16.    In sub-section "3.4. Injectability", based on Figure 4(A), the authors should explain why the extrusion forces decreased after 5mm of displacement.

17.    An important issue in this manuscript is the properties of cements in their hardness state, especially the compressive strength and porosity of cements after 24 hours of setting.

18.   The "conclusions section should be updated with the limitations of this study as well as perspectives.

Author Response

The authors wish to acknowledge the reviewer 2 for consideration the manuscript and useful comments. Fortunately, the manuscript was reformed in accordance with your comments and the other reviewers and the changes are exerted in red-font.

In more details:

Reviewer 2

R2: 1.       The abstract should be improved.

Author: The abstract was reformed.

R2: 2.      Abstract: The authors said, "The rheological measurements in oscillatory mode determined the viscoelastic nature of the pastes, in which the elastic modulus overcomes the loss modulus, especially in the BG-containing cements." There are no claims for elastic and loss moduli in the manuscript.

Author: We have examined the complementary rheological measurements including loss and storage modulus, as well as viscosity. We included them in the first version of our paper. However, in edited version, due to the increasing number of figures and the long length of the manuscript, a decision was made to eliminate the modules figures. We forget to omit the related comments from the abstract. In the reformed abstract, the exceeded comments corresponding to loss and storage modulus deleted and replaced by some explanation about viscosity.

R2: 3.      Since these bone cements are designated for orthopedic applications and more, the authors should explain which sterilization method is adequate for this type of biomaterial.

Author:  OK, the related comments was added to the last paragraph of the discussion

R2: 4.      How do the authors sterilize the samples before cell studies? This point is very important and must be highlighted.

Author:  The following statement was added in section 2.8.1.

“The samples were immersed in an antibiotic solution for a duration of 30 minutes, afterwards they were rinsed in phosphate-buffered saline (PBS), and subjected to ultraviolet irradiation (UV) for a period of 15 minutes.”

R2: 5.      The authors confirm that the dynamic frequency sweep tests have been carried out within linear viscoelastic behavior. Please use the standard nomenclature for such a parameter: linear viscoelastic region (LVR)

Author:  The sentence was reformed as follows:

“Firstly, a strain sweep mode measurement was performed to identify the linear viscoelastic region (LVR).”

R2: 6.      To homogenize the manuscript, please use elastic/viscous moduli or storage/loss moduli.

Author:  Since, in the revised manuscript, the authors focused on the viscosity, the storage and loss moduli were removed from the manuscript and only in section 2.7 when describing the equations these vocabularies were used. It should be noted that loss and storage moduli are just mentioned because they have been employed in determining complex viscosity.

R2: 7.   The authors chose to use eta’ and eta’’ instead of G’ and G’’. Why this choice? It could be preferable to use G’G’’ as a function of frequency for each sample.

Author:  In this manuscript, the authors aimed to describe the injection behavior of the cements. As you know, injectability is strongly depends on the viscosity of the cement paste. The rheological evaluation of the cement paste include measuring different characteristics such as G', G", and ɳ. In order to calculate and finally draw the complex viscosity diagrams, in oscillatory mode, we need the changes of G' and G" in frequency modes, as described in equations 2-5. The authors performed all of these tests, however as noted elsewhere, to avoid a long paper, only the viscosity diagrams are illustrated. The change in viscosity that is the main flow requirements can provide better information for understanding injection behavior.

R2: 8.      Line 171: The dynamic shear modulus is G*, not G’. Further, the equation legend must include the signification of G’ and G’’ (i.e., storage modulus and loss modulus, respectively).

Author: The description was corrected (please see section 2.7)

R2: 9.      Sub-section 2.8.1 (Cell Survival, viability, and growth): It is unclear how the cements were placed next to the cell and incubated for 24, 48, and 72 hours.

Author:  

The details were described in section 2.8.1. in brief, in the examination of MTT toxicity (Cell viability and growth), hMSC cells were first cultivated on the culture plate. Subsequently, the fine spherical specimens (with the same shape and surface area) were positioned in the cell culture medium.

R2: 10.   Figure 4A: Replace "extension" with "displacement.".

Author: Fig 4A was corrected

R2: 11.    The authors should make a difference between room temperature and 25°C.

Author: Where the authors state room temperature, they mean 25 ⁰C, because the test was set at this temperature. Thus, we correct it and mention 25°C instead of “room temperature”.

R2: 12.    The authors can add and use this bibliographic reference to highlight extrusion and the relationship between rheology and injectability for such biomaterials: https://doi.org/10.1007/s10856-012-4640-4

Author: The following reference was added (Ref No. 31).

Fatimi A, Tassin JF, Bosco J, Deterre R, Axelos MA, Weiss P. Injection of calcium phosphate pastes: prediction of injection force and comparison with experiments. Journal of Materials Science: Materials in Medicine. 2012 Jul;23:1593-603.

R2: 13.    Line 281: …the friction of the syringe surface and the paste. The authors should confirm if the existence of wall slips was checked in this case.

Author: The author provided some related explanations (in section 3.4) and the following references to support the statement:

Borhan S, Hesaraki S, Behnamghader AA, Ghasemi E. Rheological evaluations and in vitro studies of injectable bioactive glass–polycaprolactone–sodium alginate composites. Journal of Materials Science: Materials in Medicine. 2016 Sep;27:1-5. (Ref No. 29)

Medrano-David D, Lopera AM, Londoño ME, Araque-Marín P. Formulation and characterization of a new injectable bone substitute composed PVA/borax/CaCO3 and Demineralized Bone Matrix. Journal of functional biomaterials. 2021 Aug 11;12(3):46. (Ref. No. 32)

R2: 14.    Sub-Section "2.1. Initial powders preparation": the Mw of the chitosan should be specified.

Author: OK it was added.

R2: 15.    In sub-section "2.6. Injectability", the following information about the commercial 30 mL syringe should be specified: trademark, supplier, diameter, and length. In addition, there was no idea about the needle used in this study. If yes, the authors should provide the diameter and length of the needle. If not, the authors should explain why not.

Author: The required information was added. We could have used a needle at the end of the syringe, but because the length of the cannula was suitable for checking and comparing injectability (1 mm), we did not use a needle here.

R2: 16.    In sub-section "3.4. Injectability", based on Figure 4(A), the authors should explain why the extrusion forces decreased after 5mm of displacement.

Author: No filter-pressing phenomenon was observed during the injection. The paste is coherent and greasy, due the presence of chitosn. Thus, when a volume the paste is extruded, lower force is required to extrude the remaining paste. Thus, a decreased profile is seen for the extrusion by syringe displacement.

R2: 17.    An important issue in this manuscript is the properties of cements in their hardness state, especially the compressive strength and porosity of cements after 24 hours of setting.

Author: The total porosity and compressive strength measurements were determined (methods described in section 2.3) and the results are reflected in section 3.2. (Table 2).

R2: 18.   The "conclusions section should be updated with the limitations of this study as well as perspectives.

Author: The conclusion was updated with the perspectives and limitations. In addition, it was provided in a single paragraph, in accordance with the comments of reviewer 3.

Reviewer 3 Report

Title - Design and Manufacture of Bone Cements Based On Calcium Sulfate Hemihydrate and Mg, Sr-Doped Bioactive Glass

The manuscript by Moazeni et al. is interesting and requires revision for publication in the biomedicines as follows:

Comments:

1.     Abstract, the statement of novelty of this study should be stated.

2.     Lines 72-16, this information on chitosan can be elaborated with minor details properties and their derivatization of biotechnological application, i.e., https://doi.org/10.1007/s12088-018-0764-7.

3.     Introduction, many small paragraphs should be combined with corresponding paragraphs (previous).

4.     Lines 109-113, the statement of novelty/significance can be provided in this study.

5.     Figure 2, how about particle size distribution data in various synthesis conditions? Please provide.

6.     Discussion - overall, the discussion is weak, it can be significantly improved with quantitative data and justifications of findings with literature (citations) for the significance.

7.     Conclusions, this section should be in a single paragraph. Also, add a few sentences as limitations and perspectives of this study.

Minor editing of English language is required

Author Response

The authors wish to acknowledge the reviewer 3 for consideration the manuscript and useful comments. Fortunately, the manuscript was reformed in accordance with your comments and the other reviewers and the changes are exerted in red-font.

In more details:

Reviewer 3

R3: 1.     Abstract, the statement of novelty of this study should be stated.

Author: A novel composition comprising Mg, Sr-containing bioactive glass and CSH in the presence of chitosan and citric acid was developed and evaluated in terms of physical properties, strength, flow as well as some biological behaviors.

R3:  2.     Lines 72-16, this information on chitosan can be elaborated with minor details properties and their derivatization of biotechnological application, i.e., https://doi.org/10.1007/s12088-018-0764-7.

Author: The information was added using appropriate method and the requested reference was also added (Ref No. 23).

R3:  3.     Introduction, many small paragraphs should be combined with corresponding paragraphs (previous).

Author: The introduction section was deeply reformed.

R3:  4.     Lines 109-113, the statement of novelty/significance can be provided in this study.

Author: The authors designed and investigated a bone cement composition based on calcium sulfate, Mg, Sr-doped bioglass and chitosan for the first time. In the last paragraph of introduction, the novelty is stated.

R3:  5.     Figure 2, how about particle size distribution data in various synthesis conditions? Please provide.

Author:  The particle size distribution was calculated using image J software and the resultant graphs were inserted in the inset of each figure. The related comments were also provide in the text (3.3)

R3:  6.     Discussion - overall, the discussion is weak, it can be significantly improved with quantitative data and justifications of findings with literature (citations) for the significance.

Author: we tried to improve the discussion by more explanation of the results and comparing them with other similar researches. The new comments in discussion are clarified in red-font 

 R3:  7.     Conclusions, this section should be in a single paragraph. Also, add a few sentences as limitations and perspectives of this study.

Author: The conclusion section was reformed based on the reviewer's comments.

Round 2

Reviewer 2 Report

All my comments have been addressed properly, and the corrections are acceptable.

Reviewer 3 Report

Accept